# Off-Label Prescribing in Pediatric Population—Literature Review for 2012–2022

**DOI:** 10.3390/pharmaceutics15122652

**Published:** 2023-11-21

**Authors:** Valentina Petkova, Dilyana Georgieva, Milen Dimitrov, Irina Nikolova

**Affiliations:** 1Department of Social Pharmacy, Faculty of Pharmacy, Medical University of Sofia, 1000 Sofia, Bulgaria; 2Department of Pharmaceutical Technology, Faculty of Pharmacy, Medical University of Sofia, 1000 Sofia, Bulgaria; dgeorgieva@pharmfac.mu-sofia.bg (D.G.); mdimitrov@pharmfac.mu-sofia.bg (M.D.); 3Department of Pharmacology, Pharmacotherapy and Toxicology, Faculty of Pharmacy, Medical University of Sofia, 1000 Sofia, Bulgaria; inikolova@pharmfac.mu-sofia.bg

**Keywords:** off-label use, children, neonates, pediatrics, safety, prescription, legislation

## Abstract

Off-label prescribing is widespread among pediatricians, and it is unlikely that this trend will soon be bound by a uniform legal framework. This is necessitated by the fact that there are four variables: the patient’s health condition, the physician’s experience and knowledge, the legislative measures (laws, directives, guidelines, and recommendations), and finally, the pharmaceutical industry. There is considerable concern worldwide about the use of off-label medicines in children. We may call it an enormous global problem that is much talked about and written about; however, we should not forget that the goal around which everyone should unite is the patient’s life. For healthcare providers, the most important thing will always be the health and preservation of the patient’s life, particularly when it comes to children with life-threatening conditions in neonatal and pediatric intensive care units (NICU and PICU). The study aimed to examine the prevalence of off-label drug use in pediatrics. Literature research was conducted, and we included studies from 2012 to 2022 that evaluated off-label drug prevalence in various pediatric patient populations.

## 1. Introduction

Off-label use is very common and generally legal unless it violates ethical guidelines or safety regulations. Often the reason is to respond to patients’ medical needs or enable access to innovative medicines, especially when there is no alternative option [1,2]. Compared to the drugs that are authorized for adults, those licensed for pediatric use are comparatively a much smaller fraction. In addition, off-label use of medicines is in general not supported by the same level of evidence as medicines licensed for adults [3]. This may result in increased uncertainty on efficacy as well as the risk for adverse drug reactions (ADRs).

Once a medicinal product is licensed, it is the prescribers who assess the benefit–risk ratio and decide whether that medicinal product should be prescribed. All prescriptions that do not comply with the licensed Summary of Product Characteristics (SmPC), whether indication, age, dose or dosage regime, route of administration, mode of use, etc., fall under ‘off-label use’ [4]. The SmPC is a legal document, approved by national regulatory agencies, EMA, FDA, TGA, etc., as part of the marketing authorization of each medicine. However, for better or worse, ‘off-label’ is a real fact, a working phenomenon, despite not completely legitimate, that slowly paved its way into therapeutic regimens, and we already accepted it as a regular practice [5,6].

Medicines regulation controls how medicinal products are marketed, not how they are prescribed [4]. Regulatory approval is a costly and time-consuming process. It is obvious that not every drug will be tested for every eventual indication in its entirety. Thus, the regulation of therapeutic freedom adopts “an anything not explicitly prohibited is permitted” approach and assumes that medicines can be used in ways not specified in the label as long as they are prescribed by a competent professional based on scientific knowledge [4,7]. Healthcare providers are not required to limit prescriptions or recommendations to the indications approved by their country’s drug regulatory body. Despite the risk, healthcare professionals often prescribe various medications that do not contain regulatory labels for use in pediatrics. The standard of care for many conditions involves off-label uses. When this process is unavoidable, it should always be guided by a rigorous benefit/risk assessment since healthcare professionals are solely responsible for off-label use [8]. In other words, properly understanding why off-label use is common and “usually appropriate, rather than rare and usually inappropriate”, requires understanding [9].

The reason that many medications in children are administered in an off-label manner, is: (1) there are not enough legalized medicines for the pediatric population, and (2) there are not enough pharmaceutical dosage forms suitable for use in children [10]. The existence of many legal restrictions on the conduct of clinical trials in children further leads to a lag in the regulation of medicines for pediatric use, and hence the development of pediatric dosage forms suitable for both parenteral and oral use [11,12].

The current review aims to evaluate the worldwide prevalence frequency of off-label prescriptions in children in the last decade (2012–2022) and to identify the key determinants for the future aspects of enhancing the proper treatment of the pediatric population. The legislative attempts and their role are briefly explored. The current review article identifies and summarizes the existing issues for off-label treatment in the pediatric population and the recent achievements in this area.

## 2. The Need for Pediatric Drugs

### 2.1. Features of the Children

Childhood is characterized by periods of rapid growth, maturation, and development. It is widely acknowledged that children constitute a diverse and heterogeneous population, encompassing preterm neonates to post-pubertal adolescents, and are not simply miniature versions of adults [13]. The characteristics of children, in terms of physiology and development, differ from those of adults, and these also differ in the age range from newborn to adolescence [14,15]. The pharmacokinetics (PKs) and pharmacodynamics (PDs) of drugs may be altered by age and development and can be significantly affected by different factors to an extent that is not well studied to date [16,17]. In addition, the age groups of children themselves differ from each other [18].

In neonates, it is particularly difficult to detect small but significant effects as outcome measures are more difficult to assess [15]. Developmental stages can also alter the action and response to a drug. This is true for the desired action and ADRs [19]. Unfortunately, history taught us that different drug effects seen in children can be toxic, as seen with chloramphenicol, valproate, and tetracycline, or enhanced, as seen with some treatments for leukemia [20]. In addition, the natural course of disease in children may differ from that seen in adults, and they may suffer from diseases not common in adults [21]. Understanding the differences in physiology at different stages of development assists with designing drug formulations and dose regimens [22]. Age-related periods in a child’s development are defined as neonate/newborn (ages 0–29 days); infant (>28 days–12 months); toddler (>12–23 months); preschool child (2–5 years); school-aged child (6–11 years); and adolescent/teen (12–18 years).

### 2.2. Drug Formulations for Children

The development of age-appropriate dosage forms and strengths is a major challenge for the pharmaceutical industry [23]. Our knowledge of child-appropriate formulations has many gaps, and the industry faces many challenges in the process of creating appropriate formulations for all age groups [24,25]. The availability of suitable dosage forms is also limited even when the drug is approved for use by children. The complexity comes from the fact that accurate dosing is needed for different age groups [26]. It is unlikely that a single formulation will be appropriate across the pediatric population, necessitating multiple product variants [27]. Pediatric dosage forms must also be adapted to the chemical–pharmaceutical properties of the active substance and excipients, taste masking, the quantity taken, acceptability, and convenience to the patient as well as to caregivers [28]. The excipients used in pediatric formulations need to be appropriate for the age group [29,30] to avoid the consequences of excipient toxicity [31]. All these factors further complicate the manufacturing process.

The oral route of administration remains the most preferred due to its convenience and stability [32]. Recent advances in pharmaceutical technology led to the development of various types of tablets, such as melts [33], chewable and orodispersible tablets, oral lyophilizates and oral films, powders, granules [34], and pellets or sprinkles for reconstitution; however, generally, liquid products are preferred, especially in children under 6 years old [35]. The major barrier in the development of oral liquid formulations is the taste-masking of drugs, a hurdle that can be very costly and may not be totally achievable [36].

Drug acceptability is of great importance for receiving adequate therapy [37]. Pediatric formulations must be appropriate for the child to ensure compliance with the medication [28]. Different pharmaceutical forms may differ in their pharmacokinetic profile, highlighting the risks associated with the use of drugs that are not intended for use in children or were manipulated. The manipulation or extemporaneous preparation of medicines by pharmacists has its risks due to the lack of sufficient data on the quality of the final product [38,39]. Frequently, additional adjustments to medications are made by parents and caregivers to enhance adherence, especially in the pediatric demographic, where 19% of administered medications undergo manipulation [40,41].

Due to the diverse factors that influence the development of pediatric dosage forms, not forgetting the high manufacturing cost, it is lagging in pace compared to the development of adult dosage forms [42].

### 2.3. Dosing in Children

Drug dosage determination is challenging in children since traditional pharmacokinetic studies are difficult to conduct and are subject to a greater number of ethical considerations [22]. Modifications related to typical growth and development necessitate the use of evidence-based approaches to determine safe and efficient medication doses for children at various developmental stages. Additionally, it is crucial to design suitable delivery systems for these medications. Due to the lack of sufficient studies related to PKs in children, it is still a common practice to calculate pediatric doses from data obtained from the adult population [27]. However, dosage adjustments are often more complex than simply reducing the dose set for adults based on the child’s age or weight [43].

The US Food and Drug Administration (FDA) and the European Medicines Agency (EMA) emphasized the absence of pediatric clinical trials and dosing details as critical clinical gaps. There is now a demand for increased pediatric data in the assessment of new drugs [44]. Recently, Shaniv et al. [45] identified eight different neonatal formularies worldwide (Europe, the USA, Australia-New Zealand, Middle East), and six of them were compared. Each formulary varies in style and monograph template, drug information, dosing information, and update routine. Healthcare professionals may retrieve required drug dose details; however, institutions usually give access to only one formulary (if any), thus limiting the amount of essential information. For example, Suwa et al. [46] compared 72 products with pediatric indications in the USA and found that only 83% (60/72) and 43% (32/72) of the products had pediatric indications in the UK and Japan, respectively.

Most pediatric drug dosing is usually based on weight (mg/kg) or body surface area (BSA; mg/cm^2^) [47]. Utilizing total body weight (TBW) is a widespread and suitable method for calculating drug doses in children. Nevertheless, it is important to note that pediatric drug dosages cannot be directly standardized from an adult dose based solely on TBW (i.e., 60 kg adult is not equal to 60 kg child) when pediatric dosing data are unavailable [48]. Additionally of note is a very pressing issue in the last 15–20 years about drug dosing in obese children [49]. In obese children, dosing based on body weight and BSA might result in doses exceeding the maximum recommended for adults. To address this, alternative weight measures, such as ideal body weight (IBW) and adjusted body weight (ABW), were devised to better accommodate these variations. This is because the volume of distribution (Vd) and clearance (Cl) are mostly affected by physiological changes (body mass, extracellular water, tissue perfusion, and proportions of lean and fat tissue) that occur during childhood development [50,51]. Generally, hydrophilic active substances should be dosed on IBW, partly lipophilic on ABW, and lipophilic on TBW [52]. Collectively, the physiological and pharmacokinetic changes in children with obesity may require adjustments to the loading and maintenance dose, dose interval, and time to reach a steady state in certain medications—a severely hampered process when it comes to off-label use [53].

### 2.4. Conducting Clinical Research with Children

Well-designed controlled clinical trials provide reliable evidence of treatment efficacy through rigorously controlled testing of human interventions. Conducting pediatric trials is challenging due to the heterogeneity of the population and specific ethical issues [54,55]. Ethical concerns about the inclusion of children in clinical trials are disproportionately high, resulting in strict laws and ethical guidelines [56,57]. The safeguarding of children traces back to the Nüremberg code, which permits clinical investigations only in individuals capable of giving informed consent [58], followed by the “Belmont Report” [59], ICH Guideline For GCP [60] and ICH Topic E11 [61], Directive 2001/20/EC, Ref. [62], and others. In the USA, the National Institute of Health (NIH) considers, under a legal framework, the inclusion of children in clinical trials unless there are “scientific or ethical reasons to exclude them”. Furthermore, in 1998, the FDA approved a requirement that new drugs intended for use in children entering the market must undergo prior evaluation on “clinical tests in the pediatric population to determine their security, effectiveness and correct dose” [63,64].

In Canada, the revised version of the Tri-Council Policy Statement: Ethical Conduct for Research Involving Humans—TCPS 2 (2022) stipulates that the inclusion of pediatric patients in clinical trials is permissible only when the study’s objectives cannot be achieved through alternative means, informed consent is secured from parents or legal representatives, and the research poses no more than minimal risk to the children [65]. Moreover, numerous countries formulated ethical guidelines and regulatory frameworks for conducting clinical research involving children [1].

The number of clinical trials conducted in children is relatively small worldwide [66]. There exist neither mandatory requirements nor enough financial drivers for the development of pediatric medicines. The high development costs and low expected returns of new medicines for children do not usually attract the pharmaceutical industry to invest in this area. Globally, the population of children aged 0 to 14 is declining. For 2022, they comprise 25%, in comparison to 30% and 35% in 2000 and 1980, respectively. In 2022, in the EU, children under 14 years only account for 15% of the total population [67].

A high level of evidence is a prerequisite that seriously limits available drug treatment for children, as the underlying evidence is low across ages and drug classes. In a recent analysis by van der Zanden [68], findings revealed that only 14% of all off-label records (*n* = 2718) were substantiated by high-quality evidence, with 4% stemming from meta-analyses or systematic reviews and 10% from high-quality randomized controlled trials (RCTs). Furthermore, ethical, harmful, and consent concerns often pose challenges in obtaining institutional review board approval for clinical trials involving children [69]. Some researchers advocate for concurrently conducting phase I/II clinical trials for both adults and children, emphasizing the importance of interim reports from adult trials to inform and enhance clinical studies involving children [70,71,72,73]. This approach aims to minimize the time required to gather valuable and valid data on pediatric treatment while safeguarding children from exposure to ineffective and harmful treatments [74]. Consequently, the scientific community bears the responsibility of encouraging clinical trials in children to uncover novel and effective treatments. Viewing pediatric clinical research from a broader perspective, it underscores both the right of children to access efficient, evidence-based treatments and the obligation of health authorities and regulatory agencies to provide high-quality, evidence-based medical care.

## 3. Legislative and Ethical Measures for Off-Label Restriction

Over the past two decades, changes in drug regulation generated by the FDA and EMA resulted in substantial changes in how new drugs with potential use in children are studied and labeled [75,76]. To achieve child’s health protection and to ensure that medications are used ethically, in 2007, the European Union (EU) issued legislation for the development and authorization of pediatric drugs [77,78]. In addition, pharmaceutical companies are required to submit a pediatric investigation plan (PIP) to the EMA’s Pediatric Committee (PDCO) for every new medicine unless an exemption (waiver) is granted [79]. Ten years after Pediatric Regulation came into force, a total of 273 new medicines and 43 additional pharmaceutical forms appropriate for use in children were authorized in the EU [80].

Several governmental regulations were established to address off-label use in children. For example, the Pediatric Research Equity Act (PREA) of 2003 mandates pharmaceutical companies to investigate the impacts of new drugs on children when these drugs have the potential for pediatric prescriptions [81]. Studies conducted under PREA are obligatory, while those under the Best Pharmaceuticals for Children Act (BPCA) are voluntary. BPCA incentivizes pharmaceutical companies with an additional six months of patent exclusivity for drugs already on the market if they conduct clinical trials involving children [82]. These regulations not only underscore the importance of obtaining pediatric safety, efficacy, and dosing information but also contribute to the transparency of the drug approval process.

In the United States, once a drug receives approval for a specific purpose, physicians have the liberty to prescribe it for any other purpose they deem safe and effective in their professional judgment, irrespective of official FDA-approved indications [4]. The FDA lacks legal authority to regulate medical practice, allowing physicians to prescribe drugs off-label. Contrary to common belief, the use of drugs off-label is legal in the USA and many other countries. In 2014, the American Academy of Pediatrics issued a statement addressing the off-label use of pharmaceuticals in children [83]. The statement advises pediatricians that “Off-label use is neither incorrect nor investigational if based on sound scientific evidence, expert medical judgment, or published literature”. Moreover, the statement advocates for increased support and incentives for clinical testing of drugs in children and the publication of all results, regardless of positive outcomes [83]. Therefore, for doctors to be able to treat their patients safely and avoid experimental treatments, their choices should be based on scientific evidence [84]. This leads to the conclusion that for such evidence to be available, doctors and manufacturers should be incentivized to collect and publish data on off-label use. This would contribute to driving the off-label use process.

In the United Kingdom, physicians are permitted to prescribe medications off-label. In line with guidance from the General Medical Council, the physician must ensure there is ample evidence or experience supporting the medicine’s safety and efficacy. Off-label prescribing may be deemed necessary when no appropriately licensed medicine is accessible to address the patient’s requirements or when the prescription is part of approved research [85].

In China, a guideline was introduced in 2022 with the aim of aiding the management of pediatricians, pharmacists, medical managers, policymakers, and primary care physicians in handling the off-label use of drugs in pediatric cases. Additionally, the guideline offers recommendations for shaping future healthcare policies [86].

An important ethical issue in this context is determining the responsible party for informing parents or caregivers when a child is prescribed an off-label drug. Occasionally, physicians do not provide this information, while on the other hand, the pharmacist is obligated to give it [87]. Hence, two possibilities follow: (1) the parents are stressed and refuse to give their child a drug with no established efficacy and safety and (2) the parents return to the doctor in search of another alternative drug [88]. In our view, to avoid complications, it is a physician’s obligation (evidenced by giving informed consent), later confirmed by the pharmacist.

## 4. Methods

### 4.1. Search Strategy for the Prevalence of Off-Label Use

A literature search for the prevalence of off-label drug use was conducted in PubMed, Scopus, ScienceDirect, and Web of Science. Medical subject headings and free text searches were identical in all databases (“off-label use”, “prevalence”, “pediatric”, “children”, “neonates”, and/or “study”). Identified article titles and abstracts were reviewed, and articles were included if evaluating off-label drug uses, with a clear description of the health care setting and studied population. The duplicates were removed from the identified papers with the help of Zotero software (v6.0.26), and the rest were browsed for relevance. The search strategy, flow diagram, and retrieved articles are presented in Figure 1.

Only full-text research articles published in English, between January 2012 and December 2022, and reporting on the prevalence of off-label prescribing in children were included in this review. All papers considered relevant and meeting the selected criteria (*n* = 42) were original articles discussing the off-label treatment in the pediatric population.

### 4.2. Inclusion and Exclusion Criteria

Reference lists were searched to identify any relevant articles. Therapeutic and clinical guidelines, conference proceedings, book chapters, therapeutic strategies, and updates, as well as all systematic or other reviews, were excluded.

Two review authors independently conducted the quality assessment of the eligible studies. Owing to disparities in study design, children’s age, study duration, and outcome measures, statistical pooling of data was not performed. Instead, a thematic analysis of the included studies was undertaken, involving a meticulous examination and reexamination to identify emerging themes and groupings of similar themes from the articles.

Data extracted included: the country where the study was performed, a clear description of the health care setting, the studied population, the number of patients included in the study, the duration of the study, the number of prescriptions, and off-label drug prescriptions as a percent. The included articles were divided into three age groups: from 0 to 18 years old; from 0 to 15 years old; and neonates. Further, the articles were browsed for the most often off-label prescribed drug groups.

## 5. Results

### 5.1. Overall Prevalence of Off-Label Drug Prescribing in Pediatrics

We reviewed published literature from 2012 to 2022 to provide an up-to-date summary of the extent of off-label use in pediatric patients. The proportion of off-label prescriptions in our survey ranged from 3.3% (non-clinical settings; Italy) [89] to 94% (for NICU patients; Ireland) [90]. Comparing our review with some previously published literature, Moulis et al. [91] and Gore et al. [92] reported 7.0–78.7% and 36.3–97.0%, respectively. Magalhães et al. [93] retrieved 34 studies in which off-label prescriptions ranged from 12.2% to 70.6%. Balan et al. [94] reported a more extensive range of off-label prescriptions (i.e., 1.2–99.7%). Experience in several countries showed that despite it being almost impossible to estimate real-world data, especially in outpatients [95], many published retrospective and prospective trials studies estimate a high proportion of off-label pediatric prescribing [6]. As might reasonably be expected, patients treated out-of-hospital are less likely to be prescribed off-label medications [89,96,97,98]. In a hospital setting, the disease and patient’s condition are of primary importance. Oshikoya et al. [99] reported a low rate of off-label prescriptions in the chronic treatment of epilepsy, asthma, and sickle cell anemia, while in the most vulnerable group of NICU patients, the percentage skyrocketed to 94% [90]. Comparative data on the extent of off-label prescribing in children 0 to 18 years in different countries and regions, published in medical literature from 2012 to 2022, are summarized in Table 1.

In the group of studies involving children up to 15 years of age (Table 2), the differences in age and dose are the main reasons for off-label use. It is noteworthy that many drugs were in off-label use for many years [113]. The most often prescribed drugs, as off-label use, are from groups of drugs for the treatment of the respiratory system, anti-infectives, nervous system, and alimentary system. Different studies give varying reasons for off-label use, such as dosage/frequency [114,115], age [116], and formulation [117]. All the reported ranges are considerably wide. The prevalence rate of off-label medicine use in Germany, especially for children aged 3 to 6 years, is 48.7% [95]. In 1998, Conroy [118] performed a prospective study of off-label use in five European countries (United Kingdom, Sweden, Germany, Italy, and the Netherlands) and found that off-label prescriptions were 39%. Schaffer, Ref. [119] reported only 12.2% of prescriptions, based on age, to be off-label in Australia.

### 5.2. Prevalence of Off-Label Drug Prescribing in Neonates

Generally, neonates receive more prescriptions (mean = 5–7) per patient [117,125,126]. For instance, Langerová et al. [97] reported the lowest rate of off-label use (9.019%; *n* = 4282 patients) in children 0 to 15 years; however, the data on neonates are 40.9%, just as the other reports for this age group (Table 3).

Gore et al. [92], Cuzzolin and Agostino [131], Gonçalves and Heineck [121], and Aamir et al. [137] also reported a higher percentage of off-label prescriptions in pediatric intensive care units (PICU), pediatric medical and surgical wards, and neonatal intensive care units (NICU). In the study of Gore et al. [92], the number of patients studied ranged from 34 in NICU to 355,409 hospitalized children. Gonçalves and Heineck [121] found that 95.5% of newborns were prescribed at least 1 off-label drug, and the studied newborns were admitted to the NICU of a university hospital in Brazil. The condition’s severity and the need for intensive care in the NICU may explain the considerable number of prescriptions per patient [121,134]. Lindell-Osuagwu et al. [139] reported that in Finland, the proportion of off-label prescriptions increased from 50% in the 2001 [139] to 71% in 2011 [100]. In Mumbai, India, Chauthankar et al. [136] reported that 38% of neonates received at least one off-label drug, and overall off-label use was 12.3%. Similarly, in Brazil, Carvalho et al. [132] reported 27.7%, while Vieira et al. [134] reported 79% of off-label use. The great difference is not due to the year the study was performed, but rather to the definition of off-label, gestational age, inclusion and exclusion criteria, severity of medical condition, and variety of drugs used. This indicates that both the low and the high percentages may be accepted as valid. In contrast, a study performed in Norway [101] reported no difference between neonates (42%) and total off-label prescriptions (44%; 0–17 years).

An aspect in which we found notable differences between studies is the reason why the drug was used in off-label conditions. Such differences may be due to the definition of off-label use and varying reasons for off-label use (Table 4).

### 5.3. Prevalence of Off-Label Drug Prescribing by Drug Groups

Prevalence of off-label prescribing also varies widely with the class of drugs: off-label use was highest among anti-infectives [99,106,108,111,114,116,122,126,128,129,130,134,135,136,138], CNS drug group [99,102,106,108,109,111,114,122,126,133,134,135,136], respiratory [104,107,108,111,112,115,121,122,133], alimentary tract medicines [103,104,106,107,108,111,122,126,128,134], and cardiovascular medications [107,113,119,126,133,134]. This order was also observed in other studies [89,91,96,140,141,142], while the lowest off-label use was noted in antidiabetic drugs [143].

Four of the studies reported the dose, age, and indication as major contributors to off-label prescribing [99,106,111,112]. It is not possible to isolate individual, most commonly prescribed off-label medications, as studies vary widely from one another. Most recently, Oishi et al. [144] reported a surprisingly high (18.8%) level of off-label prescriptions of asthma inhalers in pediatric patients aged 0–14 years in Japan. Despite the fact that Carnovale et al. [89] reported that 12.8% of off-label prescriptions are contraindicated drugs, any of the above-mentioned drug classes have well-established use in protocols, clinical trials, and meta-analyses, but were not investigated in controlled clinical trials that meet regulatory agency’s criteria [132].

Different studies give varying reasons for off-label use, such as dosage/frequency [114,115], age [116], and formulation [117]. Likewise, in some studies [106,108], the existence of published scientific evidence was analyzed as well. Smeets et al. [145] described that only 14% of all off-label records (*n* = 2718) were supported by high-quality evidence based on randomized clinical trials. This indicates that data should be presented by age-related periods in a child’s development, from neonates to adolescents, rather than overall, as well as those inpatients, ED patients, and outpatients must be considered separately. The benefit of these reviews is that it can be noted that most off-label prescriptions were prescribed in neonates/infants, and especially in emergency settings [146], which is most likely due to the extreme vulnerability of this group of patients and the lack of clinical studies, including pharmacokinetic studies. These findings are supported by an EU report [79] and corroborated by systematic reviews by Balan et al. [147] and Allen et al. [6]. By providing fresh insights into the practice of off-label drug prescribing, these findings aim to contribute to the generation of novel and more effective knowledge, addressing the demand for high-quality drugs that are both safe and efficacious in the pediatric population.

In NICU, the most often off-label prescribed drug groups were anti-infectives (such as amikacin, gentamicin, vancomycin, meropenem, cefepime, and cefazoline), followed by drugs acting on the nervous system, the respiratory system, gastrointestinal system, and cardiovascular agents in [128,129,130,131,132,133,134,135,136,138].

A comparison between included studies, regarding the most commonly off-label prescribed active principles and the reason for it, was not performed in our study, due to the sheer variety of drugs used and polypharmacy, especially where NICU or PICU are concerned, different study periods, and the broad definition of off-label use, i.e., anything beyond the official approved SmPC (mode of use, age group, indication, dose and dosage regime, route of administration, contraindications, and precautions/warnings). The main variations in the studies are due to: (1) the broad definition of off-label use, which includes all the information in the SmPC, such as dose and dose regimen, including lower or higher dose, patient age and weight, route of administration, indications, and precautions and contraindications [104,111,148]; (2) the drug information included in the SmPC of the specific product or other products with the same active ingredient, as well as the use of databases and formularies (national and international) when looking for product information; (3) the patient’s condition (acute or chronic), as well as the duration of the disease; (4) the physician’s experience and the availability of national therapeutic recommendations for the disease; (5) the season in which the study was conducted; and (6) inpatient or ambulatory treatment. All these factors are equal in severity.

## 6. Discussion

In a recent collaborative position statement, the European Academy of Pediatrics and the European Society for Developmental Perinatal and Pediatric Pharmacology recommend considering off-label prescribing for children as rational and clinically appropriate, provided that the benefits outweigh the risks [149,150]. However, there is a lack of specific guidance on how to evaluate the benefits and risks of off-label use. The application of the Grading of Recommendations Assessment, Development, and Evaluation (GRADE) methodology [151], commonly used in evidence-based medicine for assessing the benefits and risks of an intervention, has limitations in this context. Crucial areas that are not adequately addressed include appropriate dose selection, evaluation of the availability of suitable drug formulations, and considerations of safety [152,153]. Special emphasis on dose selection is crucial, as detailed in Section 2, considering that age-specific alterations in pharmacokinetics (PKs) and pharmacodynamics (PDs) could significantly influence the required dose to achieve the target exposure [50,154]. In addressing the inherent challenges of off-label prescribing in children, a practical framework named Benefit and Risk Assessment for Off-label Use (BRAvO) was introduced in 2021 for healthcare professionals and guideline working groups. This framework provides guidance on whether and how to conduct a benefit–risk analysis for off-label pediatric prescribing, encompassing dose selection to ultimately enhance both drug efficacy and safety [150]. The conclusions regarding the balance and acceptance of benefits and risks are, therefore, subject to the judgment of healthcare professionals. While a standardized scoring system could objectify the assessment outcome, it may also compromise ease of use. The framework, by ensuring a structured and documented approach, promotes transparency and verifiability in the decision-making process. Additionally, it is crucial to recognize that a benefit–risk assessment is an ongoing and dynamic process. As new insights surface, the equilibrium between benefits and risks may undergo significant shifts, potentially leading to different conclusions [150]. Moreover, drug information is continually evolving with the emergence of new formulations as technology advances. This dynamic nature means that studies published at different times might have varying definitions of off-label prescription, even for the same drug.

Over the past two decades, there were significant advances in policy and legislation that support the development of medicines used from neonates to adolescents [155,156,157]. Since the introduction of the US and European legislation, we witnessed the most substantial advances in pediatric drug development, resulting in the incorporation of pediatric use labeling information in almost 700 product labels [158,159]. The Pediatric Regulation was enacted in the European Union (EU) on 26 January 2007, with the primary goal of enhancing the health of children in Europe by simplifying the development and accessibility of medicines for individuals aged 0 to 17 years. In October 2017, the European Commission released a comprehensive ten-year report on the implementation of the Pediatric Regulation [160]. The report indicates a notable increase in the availability of medicines for children across various therapeutic areas over the past decade, particularly in rheumatology and infectious diseases. However, it also highlights limited progress in conditions exclusively affecting children or those exhibiting biological distinctions between adults and children, especially in the case of rare diseases. Subsequently, the European Commission, along with the European Medicines Agency (EMA) and its Pediatric Committee (PDCO), devised an action plan to enhance the implementation of the Regulation [161].

The longstanding use of off-label medications in children is a common global phenomenon and remains a concern, persisting despite heightened awareness and enacted legislation. It is noteworthy that many drugs were in off-label use for many years [113]. A typical example is paracetamol, which is often mentioned as off-label use [96,102,109,111,118,122,124,127,129,132,136,137], despite it being on the market for nearly 70 years. The lack of pediatric information or if it is insufficient or unclear in the SmPCs emerges as one of the main factors for classifying a drug as off-label in many cases. Our literature review demonstrates that off-label use of medications in pediatric patients is a common practice accounting for inpatients and outpatients. We evaluated the problems and achievements in the off-label treatment of children. A wide range of results were detected between studies. The number of products identified varies greatly between the studies, depending on the country, the studied period (from one day [106] to two years [134]), and the number and kind of units included. Such difference may be due to the inequalities in drug therapies dependent on geographic areas and updates in SmPC. This may lead to some of the off-label uses being considered as such, because despite the experience with a drug, it is introduced in the clinical guidelines but not in the drug’s SmPC. Hence, off-label use could be associated with increased safety concerns, and under or overdosing due to insufficient or unclear information for use in children [92,162,163]. Despite various international regulatory initiatives and achievements over the past 20 years, many challenges remain in the development and evaluation of the safety and efficacy of pediatric medicines [164]. The review showed the complexity of the matter. More age-specific research is needed to provide adequate drug safety and efficacy for children. Until more data are provided, clinical decision-making should be guided by the best available scientific evidence and closely monitored while authorities, academics, and manufacturers collaborate to foster trials and authorize adequate drugs for this population. Therefore, we agree with Yamashiro et al. [165], van der Zanden et al., [150], and Kaguelidou et al. [164] that the most important thing is the compliance with the off-label prescription algorithm and the creation of a professional network of off-label prescriptions and their implications so that healthcare providers have a foundation to stand on when deciding whether to prescribe an off-label drug or not.

The problem of pediatric clinical trials existed for many years. Although there were great achievements in the last two decades, clinical trials in children will always lag behind those in adults because of the enormous diversity and specificity of this category of patients [164]. Off-label use in pediatrics is ubiquitous and primarily based on physicians’ experience, collaboration, knowledge, and existing published scientific literature.

The key determinants and interactions between the patient’s state of health, the doctor’s knowledge and experience, legislative measures, and the influence of the pharmaceutical industry on the prescription of off-label drugs in pediatrics are, first of all, the different age groups in children, characterized by their anatomical, morphological, and the psychological condition, that makes the creation of medicines for children a great challenge for the industry. Secondly, it is evidence-based medicine, which allows the medical practitioners, based on their experience, to prescribe a drug that is not intended, but which they are afraid to promote because of the risk of a possible error. The next determinant is the various legislative measures and initiatives around the world that seek to facilitate children’s access to medicines, but which are not uniform, which in turn leads to confusion in an industry characterized by globalization rather than segmentation, and this versatility leads to choosing the easiest step—avoiding development of medicines for children. Conversely, variations in regulatory criteria among national agencies can contribute to the prevalence of off-label prescriptions. Overly stringent criteria may result in reduced access to medications [166]. Given the prevalence of off-label drug use, the cooperation of health and regulatory authorities on one hand, and the pharmaceutical industry on the other side, is integral to instituting individual measures to provide safe and comprehensive pharmacotherapy for pediatric patients. Enforcement of legislation in the drug development process, along with subsequent pharmacovigilance, has the potential to enhance the quality of information and increase accountability within the pharmaceutical industry. This, in turn, could aid and streamline drug research in children [92]. Where it is common and evidence-based, the marketing authorization holder and the relevant regulatory authorities should have a shared responsibility to take appropriate measures to address legal uncertainties and safety concerns, including updating the SmPC. Health authorities and health insurance should support and thus reimburse therapeutic practices that are evidence-based or recommended by a respected and responsible professional body, regardless of labeling status. Legislation should be enacted to effectively encourage research into off-label medicines and facilitate the registration of off-label uses with a favorable balance between benefits and harms [149].

To summarize, the evidence, as shown in our study, is that different countries have different extents of off-label prescriptions (from 3.3% for non-clinical settings to 94% for NICU patients). A major problem arises that depending on the clinical setting and the region, different drugs may be prioritized for various periods, which is especially certain for antibiotics. The lack of pediatric information or if it is insufficient or unclear in the SmPC emerges as one of the main factors for classifying a drug as off-label in many of the cases. We posit that the considerable variability in the observed ranges stems from significant heterogeneity among the studies under review, likely attributed to differences in methodology and the diverse patient populations studied (ranging from general ward to ICU patients). The varying perspectives among prescribers, influenced by their experience and the availability of drugs in their local contexts, could also contribute to this heterogeneity. Notably, ICU patients, being generally more unwell, may receive a higher volume of prescriptions compared to patients in general wards. Additionally, hospitalized patients present a spectrum of diagnoses, leading to diverse drug prescriptions with variations in frequencies, formulations, dosages, and routes of administration. Because significantly more off-label medicines are prescribed at NICU and PICU, it is of paramount importance to provide the best possible information to prescribers on the appropriate use of medicines for children, especially concerning age and dose.

## 7. Conclusions

### 7.1. Achievements

In conclusion, though much progress in pediatric drug development was made over the past several years, further efforts are necessary to improve the availability of pediatric medications. Unless there is an increase in the availability of sufficient pediatric clinical trials and licensed drugs, the practice of off-label drug prescriptions should not be discouraged, but rather encouraged in a systematic manner. Our review offers recommendations for expanding scientific knowledge and understanding of the off-label use. The results are not directly comparable but provide valuable information on the issue. To allow results comparability, the need to unify the presentation of clinical studies on the prevalence of off-label use in pediatrics is highlighted. Equally, prescribers should (1) refer to authorized drug databases/formularies or drug information in the Summary of Product Characteristics (SmPC) before prescribing; (2) contemplate prescribing an off-label drug only when essential, following discussions with the patients’ parents or caregivers regarding associated risks and benefits; (3) inform the parents or caregivers and sign a consent form; and (4) closely monitor the patient, taking necessary steps to address adverse reactions following an off-label drug prescription, and ensure proper documentation while submitting the report to the country’s drug authority. In our opinion, off-label use should be accepted as an already established practice, but on the assumption that it is based on evidence-based medicine, especially when it comes to old medicines that are of particular importance for young patients.

### 7.2. Limitations

Our study had several limitations. We do not distinguish between in- and outpatients, age groups, and active principles used because of the vast divergencies (study design methodologies, terms of participants, and different study periods) between reviewed studies. We do not comment on the most often used medicines as well as the consequences of off-label prescriptions, as the present paper is a narrative review of the literature on the prevalence of off-label drug use in children for the period of 2012 to 2022.

## Figures and Tables

**Figure 1 pharmaceutics-15-02652-f001:**
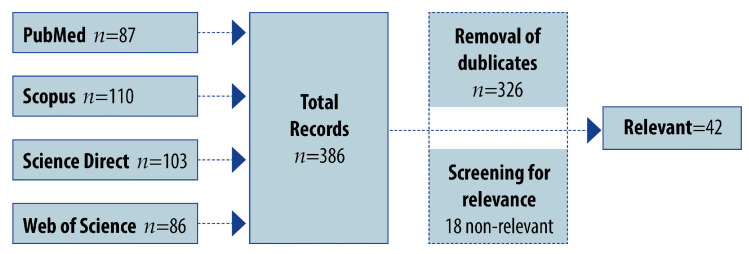
Flow diagram of the search strategy.

**Table 1 pharmaceutics-15-02652-t001:** Data on off-label prescribing in children 0 to 18 years by countries.

Country	% Off-Label Prescriptions (Total Prescriptions)	Type of Study (Number of Patients and Age)	Studied Period
Italy [89]	3.3% (4,027,119)	Retrospective *(n* = 1,708,755 non-clinical setting; 0–18 years)	12 months (January–December 2011)
Finland [100]	71% (1054)	Retrospective (*n* = 123; inpatients; 0–18 years)	0.5 months (April–May 2011)
Norway [101]	44% (930)	Prospective cross-sectional (*n* = 400 inpatients; 0–17 years)	6 months (September–October 2013 and (September–December 2014)
Sweden [102]	41% (11 294)	Prospective (*n* = 2947 inpatients; 0–18 years)	4 days (2 days in May and 2 days in October 2008)
Portugal [96]	28.1% (724)	Retrospective descriptive (*n* = 700 outpatient; 0–18 years)	12 months (January–October 2010)
Croatia [103]	19.7% (1643)	Prospective (*n* = 531 inpatients; 0–18 years)	12 months (May 2010–April 2011)
Slovakia [104]	15.7% (267)	Prospective (*n* = 68 inpatients; 2–18 years)	1 month (February–March 2011)
USA [105]	36% (1090)	Retrospective cohort (*n* = 82 inpatients; 0–18 years)	3 months (June 2008–August 2008)
Canada [106]	38.2% (2145)	Retrospective cross-sectional (*n* = 308 inpatients; 0–18 years)	A day (5 March 2014)
Australia [107]	25.7% (2654)	Retrospective (*n* = 699 ED *, in- and outpatients; 0–18 years)	24 months (January–December 2008)
Australia [108]	30.5% (6786)	Retrospective observational (*n* = 3343 ED patients; 0–17 years)	12 months (July 2011–June 2012)
Australia [109]	53.9% (1160)	Retrospective cross-sectional study (*n* = 190 inpatients; 0–18 years)	1 month (June 2013)
Malaysia [110]	34.1% (1295)	Prospective (*n* = 194 inpatients; 28 days–18 years)	3 months (April–June 2012)
Indonesia [111]	71.5% (1961)	Retrospective cross-sectional (*n* = 200 inpatients; 1 month–18 years)	3 months (August–October 2014)
Malaysia [112]	35.6% (508)	Prospective (*n* = 220 outpatients; 1 month–17 years)	6 months (July 2011–December 2011)
Nigeria [99]	7.7% (1746)	Retrospective descriptive (*n* = 477 inpatients; 0–16 years)	12 months (January–October 2015)

* ED = emergency department.

**Table 2 pharmaceutics-15-02652-t002:** Data on off-label prescribing in children 0 to ≤15 years by countries.

Country	% Off-Label Prescriptions (Total Prescriptions)	Type of Study (Number of Patients and Age)	Studied Period
Czech Republic [97]	9.02% (8559)	Prospective observational (*n* = 4282 outpatients; 0–15 years)	6 months (January–June 2012)
Slovakia [104]	21% (206)	Prospective (*n* = 49 inpatients; 0–6 years)	1 month (February–March 2011)
USA [120]	57% (240)	Prospective observational (*n* = 40 inpatients; 3 weeks–15 years)	4.5 months (November 2011–April 2012)
Brazil [121]	31.7% (731)	Retrospective cross-sectional (*n* = 705 outpatients; 0–12 years)	5 months (August–December 2012
Brazil [122]	39% (342)	Prospective cross-sectional (*n* = 342 inpatients; 0–14 years)	3 months (November 2007–January 2008)
Brazil [123]	77.8% (1158)	Prospective (*n* = 320 inpatients; 2–14 years)	6 months (September 2012–February 2013
Brazil [117]	45% (1328)	Prospective observational (*n* = 157 inpatients; 1 month–12 years)	A week period Phase 1 (August 2014) Phase 2 (January 2015)
Australia [114]	31.8% (887)	Retrospective (*n* = 300 inpatients; 0–12 years) Group 1—150 consecutive patients Group 2—150 consecutive patients	2 months Group 1 (1 July 2009–5 August 2009). Group 2 (1 January 2010–22 February 2010)
India [113]	41.25% (1789)	Prospective observational (*n* = 482 PICU patients; 1 month–12 years)	12 months (April 2012–March 2013)
India [98]	10.1% (405)	Prospective cross-sectional (*n* = 170 outpatients; 15 days–12 years)	2 months (July 2012–August 2012)
Indonesia [116]	18.6% (4936)	Retrospective population-based (*n* = 4936 outpatients; 0–5 years)	12 months (January–December 2012)
Pakistan [124]	48.6% (3168)	Prospective, observational (*n* = 895 inpatients; 1 month–15 years)	12 months (March 2014–February 2015)
Malta [115]	51.7% (209)	Prospective (*n* = 209 outpatients; 0–14 years)	A month period Phase 1 (September 2006) Phase 2 (January 2007)

**Table 3 pharmaceutics-15-02652-t003:** Data on off-label prescribing in neonates by countries.

Country	% Off-Label Prescriptions (Total Prescriptions)	Type of Study (Number of Patients and Age)	Studied Period
Spain [127]	22.5% (564)	A Prospective, observational study (*n* = 84 NICU * patients)	6 months (April–September 2018)
Slovakia [128]	43% (962)	Prospective cross-sectional (*n* = 202 NICU patients)	6 months (April–September 2012)
Portugal [129]	52.7% (1011)	Retrospective cross-sectional study (*n* = 218 NICU patients)	6 months (January–June 2013)
France [130]	59.5% (8891)	Prospective, observational (*n* = 910 NICU patients)	12 months (January–December 2012)
Italy [131]	59% (720)	Prospective (*n* = 220 NICU patients)	1-day survey (May–July 2014)
Ireland [90]	94% (900)	Prospective (*n* = 110 NICU patients)	2 months (February–March 2012)
Brazil [132]	27.7% (318)	Observational cohort study (*n* = 61 NICU patients)	1.5 months (July–August 2011)
Brazil [133]	49.3% (3935)	Prospective cohort study (*n* = 220 NICU patients)	12 months (August 2015–July 2016)
Brazil [134]	79.0% (16,143)	A nonconcurrent, hospital-based cohort study (*n* = 592 NICU preterm patients)	24 months (January 2016–December 2017)
India [135]	50% (568)	Prospective (*n* = 156 NICU patients)	3 months (June–August 2009)
India [136]	12.3% (2642)	Prospective descriptive (*n* = 460 NICU patients)	9 months (July 2014–March 2015)
Pakistan [137]	52.14% (3448)	Prospective, observational (*n* = 1300 inpatients)	12 months (May 2014–April 2015)
Israel [138]	64.8% (1064)	Prospective (*n* = 134 NICU patients)	2 months (December 2015–January 2016)

* NICU = neonatal intensive care unit.

**Table 4 pharmaceutics-15-02652-t004:** Most common reasons for off-label use in the NICU.

Reason for Off-Label Determination	Prevalence of the Reason (%)	Reference
Dosage	25.7–40–52–61.29	[129,136,137,138]
Indication	13.68–36	[137,138]
Frequency	32	[138]
Age	10.79–15–44.8	[130,137,138]
Route of administration	14	[138]
Combined (more than one reason)	0.11–8.12–15	[136,137,138]
Contraindication	13.8	[128]

## Data Availability

Not applicable.

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
