# Peer review of "Off-Label Prescribing in Pediatric Population—Literature Review for 2012–2022"

_pharmaceutics, 2023, doi:10.3390/pharmaceutics15122652_

Round 1

Reviewer 1 Report

Comments and Suggestions for Authors

Please read the attachment. Thank you. 

Author Response

Comment 1. Please align the manuscript left and right (justify).

Answer 1: The manuscript has been justified.

Comment 2. Introduction: please add a paragraph to introduce the outline of the manuscript.

Answer 2: The outline of the manuscript is presented in Section 4.

Comment 3. Please provide a flowchart for the study process.

Answer 3: The flowchart is presented in Section 4.1. (Figure 1).

Comment 4. Table 1: please make it into one page only.

Answer 4: Table 1 has been revised and transformed into a sentence.

Comment 5. Figures: their figure titles should be below their figures.

Answer 5: The title of Figure 1 has been removed to below the Figure.

Comment 6. Sections 4-5 should be combined.

Answer 6: Sections 4 and 5 have been merged.

Comment 7. Please format the manuscript following the guidance of the journal template.

Answer 7: The manuscript has been revised and formatted following the journal template.

Constructive Questions:

Question 1. What are the key determinants and interactions between the patient's health condition, physician's knowledge and experience, legislative measures, and pharmaceutical industry influence on off-label drug prescribing in pediatrics, and how do these factors collectively contribute to the prevalence of this practice?

Answer 1: We implemented the following text in Section 6. Discussion.

The key determinants and interactions between the patient's state of health, the doctor's knowledge and experience, legislative measures, and the influence of the pharmaceutical industry on the prescription of off-label drugs in pediatrics are, first of all, the different age groups in children, characterized by their anatomical, morphological, psychological condition that makes the creation of medicines for children a great challenge for the industry. Secondly, it is evidence-based medicine, which allows the medical practitioners, based on their experience, to prescribe a drug that is not intended, but which they are afraid to promote because of the risk of a possible error. The next determinant is the various legislative measures and initiatives around the world that seek to facilitate children's access to medicines, but which are not uniform, which in turn leads to confusion in an industry characterized by globalization rather than segmentation, and this versatility leads to choosing the easiest step - avoiding development of medicines for children. On the other hand, regulatory criteria may differ between national agencies, causing a prevalence of off-label prescriptions. Excessively strict criteria may cause less access to drugs. Given the prevalence of off-label drug use, the cooperation of health and regulatory authorities, parents and caregivers, and the pharmaceutical industry is integral to instituting individual measures to avoid exposing children to unnecessary risks and depriving them of potentially effective pharmacotherapy. Enforcement of legislation in the drug development process and subsequent pharmacovigilance could improve the quality of information and accountability of the pharmaceutical industry to support and facilitate drug research in children.

Question 2. How can the healthcare community and regulatory authorities work collaboratively to establish a uniform legal framework that addresses the challenges associated with off-label drug use in pediatrics while prioritizing patient safety and the preservation of children's lives in neonatal and pediatric intensive care units (NICU and PICU)?

Answer 2: We implemented the following text in Section 6. Discussion.

Where it is common and evidence-based, the marketing authorization holder and the relevant regulatory authorities should have a shared responsibility to take appropriate measures to address legal uncertainties and safety concerns, including updating the SmPC. Health authorities and health insurance should support and thus reimburse therapeutic practices that are evidence-based or recommended by a respected and responsible professional body, regardless of labeling status. Legislation should be enacted to effectively encourage research into off-label medicines and facilitate the registration of off-label uses with a favorable balance between benefits and harms.

It is a well-known fact that off-label use is based on evidence-based medicine for the sake of the patient's life and has been practiced all over the world for decades.  However, despite having a lot of accumulated clinical experience, pharmaceutical companies are rigid about the cumbersome procedure of legalizing SmPC changes. One of the possible solutions is the creation of an easily accessible database, as well as the standardization of information in the SPC, where significant differences in the level of details exist, and the available databases containing drug information in pediatric practice. This is particularly important for old products and regulatory initiatives are likely to have less impact but are of paramount importance for children. A typical example is paracetamol, which is often mentioned as off-label use, despite it being on the market for nearly 70 years. The lack of pediatric information or if it is insufficient or unclear in the SmPC emerges as one of the main factors for classifying a drug as off-label in many of the cases.

A major problem arises that depending on the clinical setting and the region, different drugs may be prioritized for various periods which is especially certain for antibiotics.

This necessitates the initiation of the procedure by the doctors with their experience and knowledge based on published data from real-world evidence-based medicine, as well as regular revisions at the hospital, national, and international levels. Because significantly more off-label medicines are prescribed at NICU and PICU, it is of paramount importance to best possibly inform prescribers on the appropriate use of medicines for children.

Reviewer 2 Report

Comments and Suggestions for Authors

COMMENTS TO THE AUTHORS

Reference: Manuscript ID: pharmaceutics-2649966

It is highlighted in blue and italics, the parts of the text that are exposed in a literal way.

The paper titled Off-label prescribing in pediatric population – literature review for 2012-2022, addresses an interesting topic.

The paper is very interesting. However, I believe that it should be taken up again by the authors and present a more elaborate, better-organized work, and also present it in the format of a scientific article.

In lines 65-69, the authors show what they do in this article, as follows:

The current review aimed to evaluate the frequency of off-label prescriptions in children in the last decade and to identify the future aspects of enhancing the proper treatment of the pediatric population. The legislative attempts and their role are briefly explored. The current article identifies and summarizes the existing issues for off-label treatment in the pediatric population and the recent achievements in this area.

The authors carry out a non-quantitative systematic review (but do not present it in this way), carrying out a particular search, the following: Lines 270-278

4.1. Search Strategy

A literature search for off-label drug use was conducted in PubMed, Scopus, ScienceDirect, and Web of Science. Medical subject headings and free text searches were identical in all databases (“off-label use”, “prevalence”, “pediatric”, “children”, “clinical trial”, and “neonates”). Identified article titles and abstracts were reviewed, and articles were included if evaluating off-label drug uses, with a clear description of the health care setting and studied population. The duplicates were removed from the identified papers with the help of Zotero software (v. 6.0.26), and the rest were browsed for relevance. The search strategy, flow diagram, and retrieved articles are presented in Figure 1.

In my opinion, the authors should present a section called “method,” in that section, frame sections 4.1 and 4.2 and offer more information about the method.

They should present a section called Results (the section they call 4.3). In this section that, the authors call 4.3, they have condensed Results and Discussion. In my opinion, they should dissociate it into two sections, or if they finally decide to show them in a single section, they should organize the information better.

In your search, they include “clinical trial”. A clinical trial is always a tightly controlled experimental investigation in a particular sample, and therefore, not indicative of what happens in the population regarding the prescription of medications for children not expressly suitable for them. To shed light on “to evaluate the frequency of off-label prescriptions in children in the last decade,” they should choose other types of research methodologies in their selection that show what happens in pediatric practices in general, not in particular experimental investigations.

Furthermore, a clinical trial is never retrospective. Many of the selected papers include this type of research.

Given that the 42 selected papers test these types of medications not expressly recommended for children, they could comment on the results, what was concluded, for example.

On lines 455-470 they write the limitations of the article:

Limitations

Our study had several limitations. We do not distinguish between in- and outpatients, age groups, and drugs used because of the vast divergencies (study design methodologies, terms of participants, different study periods) between reviewed studies. We do not comment on the most often used medicines as well as the consequences of off-label prescriptions. The present paper is a narrative review of the literature on off-label drug use in children for the period of 2012 to 2022.

However, I believe they do not show the most relevant information. They should make other tables to show the information they have not exposed. In any case, they should indicate, at least, what phase of the clinical trial the research contained in the 42 selected studies is in.

Comments on the Quality of English Language

English revision may be required.

Author Response

Comment: The authors carry out a non-quantitative systematic review (but do not present it in this way), carrying out a particular search.

Answer:   We performed qualitative research based on published literature to establish the prevalence of off-label use in a 10-year timeframe. A quantitative synthesis could not be performed due to heterogeneity in the studies, including population, design, and definition of off-label use.

Question 1. In my opinion, the authors should present a section called “method,” in that section, frame sections 4.1 and 4.2 and offer more information about the method.

Answer 1: The Section 4 has been renamed to Methods. Sections 4.1 and 4.2 have been merged. The methods have been revised and corrected.

Question 2. They should present a section called Results (the section they call 4.3). In this section that, the authors call 4.3, they have condensed Results and Discussion. In my opinion, they should dissociate it into two sections, or if they finally decide to show them in a single section, they should organize the information better.

Answer 2: Section 4.3 has been renamed to Section 5 Results. The section has been revised and corrected for more clarification.

Question 3. In your search, they include “clinical trial”. A clinical trial is always a tightly controlled experimental investigation in a particular sample, and therefore, not indicative of what happens in the population regarding the prescription of medications for children not expressly suitable for them. To shed light on “to evaluate the frequency of off-label prescriptions in children in the last decade,” they should choose other types of research methodologies in their selection that show what happens in pediatric practices in general, not in particular experimental investigations. Furthermore, a clinical trial is never retrospective. Many of the selected papers include this type of research.

Answer 3: Thank you for this comment. We used “trial” as a synonym of “study” according to ICH GCP, Section 1.12 of ICH E6(R2, however, thanks to your comment we realize it is incorrect as “clinical trial” is not synonymous with “clinical study” as per Article 2.2(1) and Article 2.2(2) of Regulation EU/536/2014. We mentioned “trial” in keywords and once in Section 5 Results. Both were superseded by “study”. We performed the search again omitting the keyword “clinical trial”. The obtained results did not change significantly as the initial search was thoroughly performed by two independent scientists. We also performed a search with the keyword study and the results were fully satisfactory. In addition, reference lists were searched to identify any relevant articles. The main part of the found studies was observational presenting real-world evidence. Thanks to your comment we replaced the keyword “clinical trial” with “and/or study”.

The main problem with the studies is that they were conducted over different time periods, different size studies, different methodologies and definitions, different seasons, etc. We have revised the cited papers and made a summary, which we have implemented in Section 5 Results, but it should be kept in mind that these data are inconclusive. This is due to the fact that many of the cited studies were published within 5 years of being conducted and secondly because over a timeframe of 10 years period, changes may have occurred in the SmPC of the drugs. There are national and international recommendations that are to be followed when administering medication to children. 

In our opinion, off-label use should be accepted as an already established practice, but on the assumption that it is based on evidence-based medicine, especially when it comes to old medicines that are of particular importance for young patients.

Question 4. Given that the 42 selected papers test these types of medications not expressly recommended for children, they could comment on the results, what was concluded, for example.

Answer 4:A wide range of results were detected between studies. The number of products identified varies greatly between the studies, depending on the country, the studied period (from one day to one year), and the number and kind of units included. We have made a number of additions to Section 5 Results by adding the most commonly used drug groups extracted from these 42 studies.

Question 5. However, I believe they do not show the most relevant information. They should make other tables to show the information they have not exposed. In any case, they should indicate, at least, what phase of the clinical trial the research contained in the 42 selected studies is in.

Answer 5: We agree that the most relevant information is concerning medicines. Going into this study, our first idea was to identify the most used drugs to help us investigate the problem in depth. However, we found a vast difference in the identified drugs in the work process. This is mainly due to inequalities in drug therapies dependent on geographic areas and the updates of reviewed SmPCs. This may lead to some of the off-label use not being considered as such in another study, because, sometimes, the experience with a drug is introduced in the clinical guidelines but, however, this information is not reflected in the drug SPCs because it has not been collected by means of regulated research and/or by the MAH responsible for marketing it.

An aspect in which we found notable differences between studies is the reason why the drug was used in off-label conditions. Such differences may be due to the inequity of SmPC information.

In the group of studies involving children up to 15 years of age, the differences in age and dose are the main reasons for off-label use. It is noteworthy that several drugs have been in off-label use for many years. The most often prescribed drugs, as off-label use, are from groups of drugs for the treatment of the respiratory system, anti-infectives, nervous system, and alimentary system. Different studies give varying reasons for off-label use, like dosage/frequency, age, and formulation. We implemented a Table showing the most common reasons for off-label use in the NICU department. In NICU, the most often off-label prescribed drug groups were anti-infectives (like amikacin, gentamicin, vancomycin, meropenem, cefepime, and cefazoline), followed by drugs acting on the nervous system, the respiratory system, gastrointestinal system, and cardiovascular agents. Because of the heterogeneity of the published results, there is no clear picture yet of the magnitude of the problem and of the potential risks to which the neonate population is exposed within NICUs.

Reviewer 3 Report

Comments and Suggestions for Authors

The authors have reviewed published literature as a basis for discussion of the use of off-label prescribing for paediatric patients.  The review  will be very useful  for the intended audience.

Some comments on the manuscript are offered below to support  any revisions to  it that the authors might undertake to improve it before it is considered further for publication.

Overall article: The editorial office might recommend transfer of the  article to a sister publication within the mdpi group that  has more appropriate scope for this work.  The journal Pharmaceutics is appropriate to work covering the design, characterisation, evaluation of, and manufacturing technologies for  pharmaceutical dosage forms and the underpinning physical, chemical and physiological sciences that support dosage form  research.  The authors manuscript focusses on prescribing and medicines utilisation, which might better be  published in Pharmaceuticals or Pharmacy.  Authors should consider agreeing to such transfer.

Page 1 line 36-43:  discusses role of SmPC.  The discussion is pertinent to territories where SmPC is enforceable.  What happens in non-European territories?  Please consider adding comment here.

Page 3 section 2.2: do authors want to add anything in this section  (or elsewhere if  more appropriate, the manipulation of adult medicines to enable paediatric dosing?  for example see Richey, et al. International Journal of Pharmaceutics 518 (2017) 155–166 and references therein.  This is an important aspect of  further risks and issues with off-label prescribing in paediatric subjects.

Page 5 lines 227-235: are these two legislative actions EU or US or what?  Please affirm where they apply.

Page 6:  this represents the length of the apparent introductory discussion , about half the manuscript.  The literature review and its discussion only start at the ned of this page.  So it it too long?  Can the  article title be  updated to indicate it is a background update and literature review not just a literature review?

Page 7  line 300 talks of 3 groups 0-18 years, 0-15 years and neonates. Do the first two groups not significantly overlap and therefore what does that mean for the review outcome?  Or are the group described incorrect?

Author Response

Comment 1. The editorial office might recommend transfer of the  article to a sister publication within the mdpi group that  has more appropriate scope for this work.  The journal Pharmaceutics is appropriate to work covering the design, characterisation, evaluation of, and manufacturing technologies for  pharmaceutical dosage forms and the underpinning physical, chemical and physiological sciences that support dosage form  research.  The authors manuscript focusses on prescribing and medicines utilisation, which might better be  published in Pharmaceuticals or Pharmacy.  Authors should consider agreeing to such transfer.

Answer 1. We have received an invitation to publish our study in [Pharmaceutics] Special Issue: Recent Advances in Therapeutic Strategies for the Treatment of Pediatric Diseases and our concept was approved. That is the reason why we have submitted it Pharmaceutics, If a transfer is needed it can be transfer to Pediatrics for example.

Comment 2. Page 1 line 36-43:  discusses role of SmPC.  The discussion is pertinent to territories where SmPC is enforceable.  What happens in non-European territories?  Please consider adding comment here.

Answer 2. Thank you for that comment. We revised the sentence line 39-41 into: The SmPC is a legal document, approved by national regulatory agencies, EMA, FDA, TGA, etc., as part of the marketing authorization of each medicine.

Comment 3. Page 3 section 2.2: do authors want to add anything in this section  (or elsewhere if  more appropriate, the manipulation of adult medicines to enable paediatric dosing?  for example see Richey, et al. International Journal of Pharmaceutics 518 (2017) 155–166 and references therein.  This is an important aspect of further risks and issues with off-label prescribing in paediatric subjects.

Answer 3. Thank you for this comment. The subject is very important and we cited the proposed author in Section 6 Discussion. Modifications to licensed drugs (e.g. dispensing a drug in a different form, extemporaneously prepared drugs, for example crushing tablets to prepare a suspension), drugs that are licensed but the particular formulation is manufactured under a special license (e.g. an adult preparation is not suitable for use in children and a smaller dose must be formulated) fall within the domain of unlicensed drugs. In the present work, we have focused only on off-label drug use. Off-label use is defined as the use of a drug outside the scope of a drug’s approved label, that is at different doses or frequencies, in different indications, in different age groups, administrated by an alternative route, or in a formulation not approved in children.

Comment 4. Page 5 lines 227-235: are these two legislative actions EU or US or what?  Please affirm where they apply.

Answer 4. The first action is EU and PREA gives FDA the authority to require pediatric studies in certain drugs and biological products. Studies must use appropriate formulations for each age group

Comment 5. Page 6: This represents the length of the apparent introductory discussion , about half the manuscript.  The literature review and its discussion only start at the ned of this page.  So it it too long?  Can the  article title be  updated to indicate it is a background update and literature review not just a literature review?

Answer 5. Owing to the fact that the paper is a review of the published literature over a 10-year period, it does not fall under the heading of a mere research paper. The first part of the paper discusses the preconditions for off-label prescribing, as well as the legal frameworks in place around the world, and the preconditions for off-label prescribing to be a regular practice around the world. The title “Off-label prescribing in pediatric population – literature review for 2012-2022” includes the period 2012-2022. This period itself includes the meaning of the word background.  We can not include the word update as this is our first paper on the subject. We would like the title to stay as it is. 

Comment 6. Page 7  line 300 talks of 3 groups 0-18 years, 0-15 years and neonates. Do the first two groups not significantly overlap and therefore what does that mean for the review outcome?  Or are the group described incorrect?

Answer 6. Thank you for this comment. There are so many variables in the retrieved articles (age, diseases, clinical site or ambulatory, drugs used, study duration, study design, season, off-label scoring criteria) that it is tough, if not impossible, to compare them. However, our study aimed to determine the trend of off-label prescribing, following the measures adopted, and factors influencing it, as well as to identify practices and possibly which drugs are most prescribed. Due to the inequity between the articles found, we decided to divide the retrieved articles by reported studied age: 0-18 years old, 0-15 years old, and neonates because we would like to present the overall of off-label prescription preference.

Round 2

Reviewer 1 Report

Comments and Suggestions for Authors

all concerns have been solved.

Reviewer 2 Report

Comments and Suggestions for Authors

I believe that the changes made by the authors have resulted in greater clarity and quality of the research contained in the manuscript.